# Enhanced Resistance of *atnigr1* against *Pseudomonas syringae* pv. *tomato* Suggests Negative Regulation of Plant Basal Defense and Systemic Acquired Resistance by *AtNIGR1* Encoding NAD(P)-Binding Rossmann-Fold in *Arabidopsis thaliana*

**DOI:** 10.3390/antiox12050989

**Published:** 2023-04-24

**Authors:** Tiba Nazar Al Azzawi, Murtaza Khan, Bong-Gyu Mun, Sang-Uk Lee, Muhammad Imran, Adil Hussain, Nkulu Kabange Rolly, Da-Sol Lee, Sajid Ali, In-Jung Lee, Byung-Wook Yun

**Affiliations:** 1Department of Applied Biosciences, Kyungpook National University, Daegu 41566, Republic of Korea; 2Department of Horticulture and Life Sciences, Yeungnam University, Gyeongsan 38541, Republic of Korea; murtazakhan@yu.ac.kr (M.K.);; 3Biosafety Division, National Institute of Agriculture Science, Rural Development Administration, Jeonju 55365, Republic of Korea; 4Department of Entomology, Abdul Wali Khan University Mardan, Mardan 23200, Pakistan; 5Department of Southern Area of Crop Science, National Institute of Crop Science, RDA, Miryang 50424, Republic of Korea

**Keywords:** nitric oxide, plant growth and defense, pathogenic bacteria, NAD(P)-binding Rossmann-fold superfamily gene, *Arabidopsis thaliana*

## Abstract

Nitric oxide (NO) regulates several biological and physiological processes in plants. This study investigated the role of *Arabidopsis thaliana Negative Immune and Growth Regulator 1* (*AtNIGR1*), encoding an NAD(P)-binding Rossmann-fold superfamily, in the growth and immunity of *Arabidopsis thaliana*. *AtNIGR1* was pooled from the CySNO transcriptome as a NO-responsive gene. Seeds of the knockout (*atnigr1*) and overexpression plants were evaluated for their response to oxidative [(hydrogen peroxide (H_2_O_2_) and methyl viologen (MV)] or nitro-oxidative [(S-nitroso-L-cysteine (CySNO) and S-nitroso glutathione (GSNO)] stress. Results showed that the root and shoot growth of *atnigr1* (KO) and *AtNIGR1* (OE) exhibited differential phenotypic responses under oxidative and nitro-oxidative stress and normal growth conditions. To investigate the role of the target gene in plant immunity, the biotrophic bacterial pathogen *Pseudomonas syringae* pv. *tomato* DC3000 virulent (*Pst* DC3000 *vir*) was used to assess the basal defense, while the *Pst* DC3000 avirulent (*avr*B) strain was used to investigate *R*-gene-mediated resistance and systemic acquired resistance (SAR). Data revealed that *AtNIGR1* negatively regulated basal defense, *R*-gene-mediated resistance, and SAR. Furthermore, the *Arabidopsis* eFP browser indicated that the expression of *AtNIGR1* is detected in several plant organs, with the highest expression observed in germinating seeds. All results put together suggest that *AtNIGR1* could be involved in plant growth, as well as basal defense and SAR, in response to bacterial pathogens in *Arabidopsis*.

## 1. Introduction

Plants are sessile organisms and they are constantly vulnerable to biotic and abiotic stresses [1]. Unlike animals, plants lack mobile defender cells and a somatic adaptive immune system to fight pathogen infection [1,2]. Therefore, plants rely on their innate immune system to detect and transmit signals and to react to infectious pathogens [2]. The plant’s protective immunological response is divided into pathogen- or microbial-associated molecular patterns (PAPMs/MAMPs-triggered immunity (PTI)), resistance (*R*) gene or effector-triggered immunity (ETI), and systemic acquired resistance (SAR) [3]. One of the earliest adaptations toward microbial infections is the oxidative burst, which triggers the hypersensitive response (HR) and results in programmed cell death (PCD) at the infection site [4]. The HR is frequently connected to a resistance response and is mediated by interactions between the avirulence proteins of the pathogen and plant resistance proteins, as well as various signaling pathways, both indirectly and directly [1]. Plants use pattern recognition receptors (PRRs) to recognize PAMPs/MAMPs and induce PTI. PTI is considered to be inhibited by a class of pathogen-encoded effector proteins known as avirulence (avr) factors [5], which are identified by host-encoded R proteins, and impart a long-lasting and strong resistance known as resistance mediated by the *R*-gene or ETI [6].

In response to biotic and abiotic stress conditions, plants produce reactive oxygen species (ROS), reactive nitrogen species (RNS), and melatonin, which are pivotal signaling molecules [4,7,8]. ROS and RNS play a dual role in plants: when produced in optimal conditions, they act as signaling molecules; when produced excessively, they act as toxic molecules, causing oxidative damage to the plants [9,10]. Recently, the role of ROS and RNS in plant growth and development and symbiotic association and defense responses against biotic and abiotic stress conditions has been reviewed in detail [7,8,9,10,11,12]. Among RNS, NO has emerged as a core signaling molecule, with regard to the increasing interest in NO-related research in plant bioscience [13], since it was designated “Molecule of the Year in 1992” for its outstanding regulatory roles in both plants and animals [8,14]. NO regulates a variety of biological processes in plants, including plant growth and development and adaptive response mechanisms under abiotic and biotic stress conditions [15,16]. Previous reports revealed the roles of the NO-induced genes *AtCLV1*, *AtCLV3*, *AtAO3*, *AtbZIP62*, and *AtILL6* in plant growth and defense against biotic and abiotic stresses. These NO-induced genes differentially regulate plant growth and defense responses against biotic and abiotic stress conditions [1,4,17,18].

The role of NO in biological systems was also revealed through RNA-seq-mediated transcriptomic studies. The latter have reported differential gene expression patterns in plants upon CySNO (an NO donor, 1 mM) application. The associations of target NO-responsive genes with the regulation of plant defense, the abiotic stress response, hormone signaling, and other developmental processes have equally been established [19]. Among the differentially expressed genes (DEGs) reported earlier, *Arabidopsis thaliana Negative Immune and Growth Regulator 1* (*AtNIGR1*) with gene number AT1G66130 was identified among the topmost down-regulated genes (fold change –5.34). This gene belongs to NAD(P) Rossmann-fold superfamily proteins and participates in oxidoreductase activity, catalytic activity, and metabolic processes (The *Arabidopsis* Information Resource, https://www.arabidopsis.org/ (accessed on 3 January 2022)). We obtained this data from the said website on 3 January 2022.

The advent of high-throughput sequencing technologies, advances in omics-related studies, and the development of bioinformatics tools and the abundance of complex biological data have offered tremendous opportunities to shift our common understanding of plant metabolism and biological processes, as well as their associated genetic factors, under both normal and stress conditions. Several bioinformatics tools available in online data, such as string (https://string-db.org, allowing for the prediction of protein–protein interactions (accessed on 7 January 2023)) have been developed to analyze multiple information resources generated using experimental or predictive models. We obtained this data from the said website on 7 January 2023. Considering the progress recorded in this field and the easy access to the public, the use of bioinformatics in molecular biology-related studies will increase with time.

This study aimed to investigate the role of the NO-downregulated *AtNIGR1*gene in plant growth and development. To achieve that, *Arabidopsis* plants were exposed to oxidative and nitro-oxidative stress conditions. To further investigate the role of our target gene in plant basal defense, *R*-gene-mediated resistance, and SAR, *Arabidopsis* mutant lines lacking (knockout, KO) *atnigr1* and overexpressing (OE) *AtNIGR1*, along with relevant control plants were inoculated with virulent and avirulent pathogenic bacteria.

## 2. Materials and Methods 

### 2.1. Plant Materials and Growth Conditions

The Nottingham *Arabidopsis* Stock Center (http://arabidopsis.info/ (accessed on 7 January 2023)) provided seeds of the *Arabidopsis thaliana* wild-type (WT) ecotype Columbia zero (Col-0) and the mutants *atnigr1* (KO), *AtNIGR1* (OE), *atgsnor1–3*, *atcat2*, and *atsid2*. We ordered the seeds from the said center and website on 7 January 2023. The SALK number of the T-DNA insertion is SALK_064843. All of the genotypes used in this investigation had a Col-0 genetic background. For seeds, sterilization, sowing, transplantation, and collection of the samples for genotyping, plant growth and immunity-related parameter analyses were performed as previously described [17]. In brief, samples were obtained at the rosette stage (4-week-old plants) for genotyping and T-DNA insertion confirmation using PCR. To confirm genotypes of the *atnigr1* (KO) and *AtNIGR1* (OE) plants of the *AtNIGR1* (AT1G66130), gene PCR was run with gene-specific forward (F) and reverse (R) primers and F border, 35SF and gene-specific R primers (Appendix A). Relative controls were used based on their established roles in plant growth and immunity, as previously suggested [4]. The *atgsnor1–3* mutant was used as a sensitive or susceptible control for nitro-oxidative and biotic stress. GSNOR is known to play a function in a variety of plant developmental programs, as well as plant immunity [20]. It is a typical line for testing the *Arabidopsis* response under variable nitro-oxidative stress conditions. The *atcat2* mutant line was used as a sensitive control for oxidative stress conditions. In *Arabidopsis*, AtCAT2 is a class I catalase expressed in the leaves, roots, and seeds, with much higher transcript abundance than the other catalases, and contributes to the circadian and photosynthetic-type rhythm [21]. In *Arabidopsis*, *AtCAT2* is responsible for the majority of catalase activity, as knockout lines of *atcat1* and *atcat3* show a smaller reduction in leaf catalase activity than *atcat2* [4,21]. Therefore, *atcat2* mutant lines are frequently used as an oxidative stress model. The *atsid2*-knockout mutant line was used to study the salicylic acid (SA) pathway [4,22]. In *Arabidopsis*, Isochorismate Synthase 1 (ICS1) is encoded by the *Salicylic Acid Induction Deficient 2* (*AtSID2*) gene. The *atsid2*-mutant line is incapable of accumulating SA and lacks SA-dependent defensive responses.

### 2.2. Redox Stress Assay

To investigate the role of the *AtNIGR1* gene in plant growth and development under control and redox stress, the plants were exposed to control (only ½ MS media), oxidative stress (supplemented with either 2 mM H_2_O_2_ or 1 µM MV), and nitro-oxidative stress (supplemented with 0.75 mM CySNO or 0.75 mM GSNO) as described previously [4]. Before sowing, the seeds were surface-sterilized for 5 min in a 50-percent commercial bleach solution provided with 0.1-percent Triton X-100 (Sigma Aldrich, USA). After that, the seeds were rinsed three times with sterilized distilled water and stratified for 24 h at 4 °C. Furthermore, following the same author, the results for phenotypic evaluation and root and shoot lengths were collected after two weeks, with at least three replicates per treatment.

### 2.3. Pathogenic Growth, Inoculation, and Electrolyte Leakage Assay

The biotrophic bacterial pathogen *Pseudomonas syringae* pv. *tomato* DC3000 virulent (*Pst* DC3000 *vir*) was used to assess the basal defense, while the *Pst* DC3000 avirulent (*avr*B) strain was used to investigate *R*-gene-mediated resistance and SAR as described earlier [23]. For selection, the bacteria were cultivated on Luria-Bertani (LB) agar media supplemented with suitable antibiotics and incubated overnight at 28 °C. The single colony was transferred to LB broth, which was supplemented with antibiotics, and cultured at 28 °C with constant shaking overnight. The bacterial strains were inoculated at a concentration of 5 × 10^5^ colony forming units (CFU) into the abaxial side of the leaves of different plant lines, including wild-type (WT), *atgsnor1–3*, *atsid2*, and KO and OE lines of the *AtNIGR1* gene. In the case of basal defense, the samples were collected for colony counts and gene expression at 0, 1, 2, 3, and 4 days and 0, 12, 24, and 48 h post-inoculation, respectively. Moreover, some of the plants were retained in the experimental setup to observe the development of disease symptoms. For *R*-gene-mediated resistance, the samples were collected at 0, 6, 12, and 24 h post-inoculation. Meanwhile for SAR, the samples were collected from distal leaves at 0, 6, 12, and 24 h post-inoculation. Only 10 mM MgCl_2_ was used to infiltrate the control plants. For this purpose, we followed previously used methods [4].

Bacterial infection-mediated oxidative stress affects the integrity of the cell membrane, and this event is followed by ion leakage out of the cell that may lead to the activation of programmed cell death and plant growth failure. In order to quantify the *Pst* DC3000-induced cell death or membrane injury, we conducted an electrolyte leakage assay after *Pst* DC3000 *avrB* inoculation as previously described [1]. Briefly, 10 uniform leaf discs (1 cm diameter) were collected with a cork borer from different plants of each Arabidopsis genotype at 1, 2, 4, 6, 8, 12, and 24 h after inoculation with *Pst* DC3000 *avrB*. Leaf samples were then rinsed three times with deionized water in petri plates to remove surface electrolytes, and transferred to 6-well culture plates (SPL Life Sciences, Pocheon-si, Korea) and floated in 5 mL of deionized water in each well for about 30 min. The electrolyte leakage of each sample was recorded over time using a portable conductivity meter (HURIBA Twin Cond B-173, Kyoto, Japan).

### 2.4. qRT-PCR (Quantitative Real-Time PCR) Analysis

For RNA extraction, complementary DNA (cDNA) synthesis, and qRT-PCR analysis, previously described methods were used [1]. The TRI-Solution™ Reagent [(Cat. No: TS200-001, Virginia Tech Biotechnology, Lot: 337871401001)] was used to extract total RNA from leaf samples, as directed by the manufacturer’s protocol and described earlier. In brief, using the BioFACT™ RT-Kit (BioFACT™, Daejeon, Korea) and the manufacturer’s standard methodology, 1 µg of RNA was used to synthesize complementary DNA (cDNA). For gene expression analysis, cDNA was used as a template in the Eco™ real-time PCR machine (Illumina, San Diego, CA, USA), using the 2X Real-time PCR Master Mix (including SYBR Green 1 BioFACT™, Daejeon, Korea), along with 100 ng of template DNA and 10 nM of each forward and reverse primer to a final volume of 20 µL. The No Template Control (NTC), which includes simply distilled water instead of template DNA, was utilized as a negative control. The real-time PCR machine was run in a 2-step reaction that included polymerase activation at 95 °C for 15 min, denaturation at 95 °C for 5 s, and annealing and extension at 65 °C for 30 s. The data were standardized using the relative expression of *Arabidopsis Actin2*, and the total reaction cycles were 40 [17]. 

### 2.5. Functional Categorization and Gene Ontology Annotation

A gene ontology (GO) annotation search was carried out for the *AtNIGR1* gene using the TAIR GO Annotations tool (Home > Tools > Bulk Data Retrieval > GO Annotations: https://www.arabidopsis.org/tools/bulk/go/index.jsp (accessed on 7 January 2023)). We obtained this data from the said website on 7 January 2023. For this purpose, the locus identifier was searched for GO annotations using the option of whole-genome characterization. The output was observed as an HTML and submitted for functional characterization. Data were retrieved for GO Cellular Component, GO Molecular Function, and GO Biological Process.

### 2.6. Gene Structure and Domain Analysis

Gene structure and domain analyses were performed using UniProtKB (uniport.org (accessed on 7 January 2023)), where the locus identifier for the *AtNIGR1* gene is Q9C8D3. We obtained this data from the said website on 7 January 2023. Data regarding the number, type, location, and sequence of different domains were retrieved from the Family & Domains section. The 3D structure for the alpha fold was also downloaded from the UniprotKB database.

### 2.7. Identification of Plant Homologs and Phylogenetic Analysis

To identify homolog genes of *AtNIGR1* in other plant species, the TAIR Plant Homologs tool PhyloGenes was employed. The tool helped us retrieve data from gene families of PANTHER16.0. PhyloGenes (http://www.phylogenes.org/ (accessed on 7 January 2023)) and displays pre-computed phylogenetic trees of gene families, along with experimental gene function data, to facilitate the inference of unknown gene functions in plants. We obtained this data from the said website on 7 January 2023.

### 2.8. Expression Pattern of AtNIGR1 in Different Organs

The *Arabidopsis* eFP Browser 2.0 of the Bio-Analytic Resource of Plant Biology, available at http://bar.utoronto.ca/efp2/Arabidopsis/Arabidopsis_eFPBrowser2.html (accessed on 7 January 2023), was used to gather relevant information on the proposed expression of our target gene (*ANIGR1*) in different plant organs. We obtained this data from the said website on 7 January 2023.

### 2.9. In Silico Prediction of S-Nitrosylation Sites

An *in silico* analysis-based approach facilitated by computer simulations using the GPS-SNO algorithm [24] allowed us to predict *S*-nitrosylation target cysteines. To achieve that, the amino acid sequence was used for the prediction. The 3D structure was downloaded from UniProtKB as a PDB file and opened in PyMol (https://pymol.org/2/ (accessed on 7 January 2023)). We obtained this data from the said website on 7 January 2023. The 3D structure is shown as a green cartoon structure, whereas the predicted *S*-nitrosylation cysteines were shown as pink bolls.

### 2.10. Protein Interaction Analysis

The interaction of *AtNIGR1* with other proteins was predicted via SMART (http://smart.embl-heidelberg.de/(accessed on 7 January 2023)) [25,26]. We obtained this data from the said website on 7 January 2023. The protein sequence was submitted for analysis, and the results were downloaded as an SVG image. 

### 2.11. Statistical Analysis

The experiments were repeated thrice for each assay, and the representative findings are shown. The data point for media stress conditions is the mean of three replicates, with five plants pooled in each replicate, whereas the data point for the pathogenicity assay is the mean of three replicates. One-way analysis of variance (ANOVA) was used to determine the significant differences between each treatment, followed by Duncan’s multiple range test using a statistical analysis system (SAS 9.1). Microsoft Excel was used to calculate mean values, standard deviations, and standard errors. GraphPad Prism software (version 6.0, San Diego, CA, USA) was used to visualize the data.

## 3. Results

### 3.1. Root and Shoot Growth Patterns of AtNIGR1 KO and OE 

Initially, we were interested in assessing the change in the phenotype of the *atnigr1*-related Col-0 in response to oxidative and nitro-oxidative stress. To achieve that, seeds of the *atnigr1* (KO) and *AtNIGR1* (OE) lines, along with the relevant controls, were grown on ½ MS media modified with H_2_O_2_ or MV (for oxidative stress), nitro-CySNO, or GSNO (for nitro-oxidative stress). All of the genotypes used in this investigation had a Col-0 genetic background. Col-0 was used as the wild-type (WT) control, the *atgsnor1–3* mutant was used as a sensitive control for nitro-oxidative stress, and the *atcat2* mutant line was used as a sensitive control for oxidative stress conditions. As shown in panels A and B of Figure 1, *atnigr1* plants had longer roots under control conditions compared to Col-0. A similar pattern was also observed when *atnigr1* plants were grown on H_2_O_2_, CySNO, and GSNO media. Under GSNO treatment, *atnigr1* and *atcat2* exhibited similar root growth patterns. However, an opposite root growth pattern (reduced root elongation) was recorded in plants overexpressing the *AtNIGR1* gene under all tested conditions, except MV, compared to Col-0. As expected, *atgsnor1–3* did not germinate on H_2_O_2_-medium but showed a significant root-inhibition pattern on MV, CySNO, and GSNO. In addition, *atcat2* and *atnigr1* had similar root phenotypes under control, MV, and GSNO conditions, but not under H_2_O_2_ and CySNO. Furthermore, plants overexpressing *AtNIGR1* exhibited shorter roots compared to Col-0 WT under all tested conditions.

In the same way, the shoot height was higher in *atnigr1* but lower in *AtNIGR1* OE plants compared to that recorded in Col-0 WT under all growth conditions (Figure 1A,C).

### 3.2. AtNIGR1 Negatively Regulates Plant Basal Defense

Plants lacking *AtNIGR1* (*atnigr1*) or overexpressing the *AtNIGR1* gene along with the relevant controls were challenged with *Pst* DC3000 *vir* to investigate the possible involvement of the target gene in basal defense in plants. All of the genotypes used in this investigation had a Col-0 genetic background. Col-0 was used as the wild-type (WT) control, while *atgsnor1–3* and *atsid2* mutant lines were used as susceptible controls for biotic stress. The results of the pathogenesis test revealed that *atnigr1* plants exhibited an enhanced resistance toward *Pst* DC3000 *vir*, while those overexpressing *AtNIGR1* showed a highly susceptible phenotypic response similar to that observed in the susceptible controls, *atgsnor1–3* and *atsid2* (Figure 2A). For further confirmation, we evaluated the pathogenic growth in the KO and OE plants of the *AtNIGR1* gene, along with *atgsnor1–3*, and *atsid2*. No significant change in pathogenic growth was observed with either genotype at 0 days post-inoculation (dpi). However, at 1–4 dpi, *atnigr1* plants showed a significant decrease in pathogenic growth, while and *AtNIGR1* (OE) plants exhibited an opposite pattern (Figure 2B).

Generally, when plants are infected with pathogenic bacteria, salicylic acid (SA) accumulates as part of the signaling event to activate the appropriate defense mechanism. In this regard, we measured the transcript accumulation of the well-known pathogenesis-related genes *AtPR1* and *AtPR2* (SA-dependent pathway marker genes) in the *atnigr1* (KO) and *AtNIGR1* (OE) plants, as well as in the susceptible control lines. The qPCR data in panels C and D of Figure 2 show that the expression level of *AtPR1* and *AtPR2* increased over time, and the peak expression level was recorded 24 hpi in *atnigr1* (susceptible) compared to the Col-0 WT. However, both *AtPR1* and *AtPR2* showed much lower expression levels in *AtNIGR1* (OE) plants compared to that recorded in the Col-0 background. 

### 3.3. Negative Regulation of R-Gene-Mediated Resistance by the AtNIGR1 Gene

To assess the possibility for the *AtNIGR1* to be involved in the signaling network upon bacterial pathogen infection, as well as in the *R*-gene mediated resistance, we inoculated *atnigr1* (KO) and *AtNIGR1* (OE) plants along with Col-0 (WT), *atgsnor1–3*, and *atsid2*, with *Pst* DC3000 *avr*B. All of the genotypes used in this investigation had a Col-0 genetic background. Col-0 was used as the wild-type (WT) control, while *atgsnor1–3* and *atsid2* mutant lines were used as susceptible controls for biotic stress. The results show that after 6, 12, and 24 h of inoculation with *avr*B, a significant increase and decrease in the transcript accumulation of *AtPR1* and *AtPR2* were observed in the *atnigr1* (KO) and *AtNIGR1* (OE) plants compared to that in the Col-0 WT (Figure 3A,B). Furthermore, when compared to the Col-0 WT, the *atnigr1* (KO) and *AtNIGR1* (OE) plants showed lower and higher electrolyte leakage, respectively (Figure 3C). In addition, compared to Col-0 WT, the disease-susceptible controls, *atgsnor1–3* and *atsid2*, showed lower transcript accumulation of *AtPR1* and *AtPR2* genes and higher loss of electrolytes (Figure 3A–C). Collectively, these results indicate that the *AtNIGR1* gene negatively regulates resistance mediated by the *R* gene.

### 3.4. AtNIGR1 Negatively Regulates Systemic Acquired Resistance

A number of SAR-associated signals have been identified so far. These include salicylic acid and its methylated derivative MeSA, azelaic acid (AZA), and glycerol-3-phosphate dehydrogenase (G3Pdh). To determine the possible role of NO-downregulated *AtNIGR1* in SAR activation, we studied the expression of *AtPR1*, *AtPR2*, *At*G3Pdh, and *AtAZI* genes in systemic leaves. For this purpose, we inoculated *atnigr1* (KO) and *AtNIGR1* (OE), plants along with Col-0 (WT), *atgsnor1–3*, and *atsid2*, with *Pst* DC3000 *avr*B. All of the genotypes used in this investigation had a Col-0 genetic background. Col-0 was used as a wild-type (WT) control, while the *atgsnor1–3* and *atsid2* mutant lines were used as susceptible controls for biotic stress. After 6, 12, and 24 h, the transcript accumulation of *AtPR1*, *AtPR2*, *AtG3Pdh*, and *AtAZI* genes was significantly higher and lower in the *atnigr1* (KO) and *AtNIGR1* (OE) plants of the *AtNIGR1* gene as compared to that in the Col-0 WT (Figure 4A–D).

### 3.5. In Silico Characterization, Gene Ontology, Predicted Organ-Specific Expression, and Homology

Prior to conducting biological assays that attempted to investigate the role of *AtNIGR1* in plant defense and oxidative and nitro-oxidative stress, we were interested in uncovering the basic biological processes, as well as predictive molecular functions, in which *AtNIGR1* may be involved. It was interesting to see that gene ontology (GO) analysis proposed *AtNIGR1* to be involved in the chlorophyll biosynthetic process and response to light stimulus (biological process) (Appendix A). In addition, Appendix A suggests that *AtNIGR1* would be associated with nucleotide binding (molecular function). Appendix A indicates that *AtNIGR1* could be located in the cytosol and nucleus.

Furthermore, Appendix A indicates that the oxidoreductase NAD(P)-binding Rossman fold protein is encoded by many other genes in other plant species, including Musa acuminate ssp. malaccensis, rice, Triticeae, Malvaceae, etc. Of all species, *AtNIGR1* is suggested to be closer to its ortholog in Brassica.

Moreover, the prediction of organ-specific expression analysis (eFP browser, TAIR) suggests *AtNIGR1* expression in almost all shoot components of the plant to varying degrees. However, the highest expression is proposed in germinating seeds, followed by the cotyledonary leaves, vegetative rosette leaves, and rosette leaves after flowering. Significant expression was also observed in the siliques (Appendix A).

### 3.6. In Silico Prediction of Protein Interaction and S-Nitrosylation Sites 

The prediction of protein–protein interaction proposes possible interactions of *AtNIGR1* with other proteins, including the Mevalonate/galactokinase family protein GALK, Galacturonic acid kinase GalAK, arabinose kinase ARA1, putative xylulose kinase XK-2, GroES-like zinc-binding alcohol dehydrogenase, and various Glyceraldehyde-3-phosphate dehydrogenases (GAPs) (Figure 5A and Table 1). In addition, the prediction of potential *S*-nitrosylated sites allowed us to predict that the cysteine residues Cys-225 and Cys-359 may be exposed with a high probability of being *S-*nitrosylated (Figure 5B and Table 2). C225 was found to be located in the middle of Cluster B, whereas C359 was located at the end of Cluster C. 

## 4. Discussion

### 4.1. NO-Downregulated AtNIGR1 Gene Negatively Regulates Root and Shoot Lengths of A. thaliana

Nitric oxide (NO) is important in diverse cellular activities and biological processes, including plant growth and development, and responses to biotic stress [8,9,11]. Microarrays [27], RNA-seq [19], and qPCR [28] have helped reveal changes in transcript accumulation at a genome-wide level in response to NO. Under normal conditions, plants allocate their energy to the growth and development process. However, in the event of stress, plants tend to redirect their resources to trigger the defense mechanisms, which may result in delayed or reduced growth. On the one hand, our data showed that *atnigr1* plants exhibited longer roots and higher shoot heights compared to the WT under normal, oxidative, and nitro-oxidative stress conditions. On the other hand, plants overexpressing the *AtNIGR1* had shorter roots and shoot lengths than Col-0 WT (Figure 1A–C). The observed differences in root and shoot growth between the *atnigr1* (KO) and *AtNIGR1* (OE) of the *AtNIGR1* gene (Figure 1) suggest that the NO-downregulated *AtNIGR1* gene may play a role in regulating plant growth processes.

### 4.2. NO-Downregulated AtNIGR1 Gene Negatively Regulates Basal Defense, R-Gene-Mediated Resistance, and SAR of A. thaliana

Different genotypes exposed to a particular pathogen are expected to exhibit different phenotypic responses toward that pathogen [1,17]. Within the plant, a wide range of defense mechanisms and signaling networks are induced to defend the plant. Sensitive genotypes are likely to show severe damage compared to resistant ones, in part, due to the presence or activation of several defense factors that trigger the innate immune system [29,30]. Here, we inoculated the *atnigr1* (KO) plants and those overexpressing the *AtNIGR1* gene to assess their phenotypic response toward a bacterial pathogen, along with the Col-WT and the disease-susceptible mutant lines, *atgsnor1–3* and *atsid2*. Therefore, the observed enhanced resistance of the *atnigr1* (KO) line and the susceptibility of the *AtNIGR1* (OE) line (Figure 2A,B) following *Pst* DC3000 *vir* inoculation, coupled with the pathogenic growth suggests that the *AtNIGR1* gene would act as a negative regulator of plant basal defense against bacterial pathogens. The negative regulation of plant basal dense by the NO-downregulated *AtNIGR1* gene was further confirmed via the transcript accumulation of *AtPR1* and *AtPR2* genes. The results showed that after inoculating the *Pst* DC3000 *vir* bacteria, in comparison to that in the Col-0 WT, the expression of *AtPR1* and *AtPR2* genes significantly increased and decreased in the *atnigr1* (KO) and *AtNIGR1* (OE) plants, respectively (Figure 2C,D). Therefore, we could then suggest that *AtNIGR1* would be involved in the negative regulation of basal defense in *Arabidopsis*.

To explore the role of the NO-downregulated *AtNIGR1* gene in the *R*-gene-mediated resistance of *A. thaliana*, the plants were inoculated with *Pst* DC3000 *avr*B. The results showed that after inoculating the pathogenic bacteria, in comparison to that in Col-0 WT, the expression of *AtPR1* and *AtPR2* genes significantly increased and decreased in the *atnigr1* (KO) and *AtNIGR1* (OE) plants, respectively (Figure 3A,B). Furthermore, electrolyte leakage experiments showed that the *atnigr1* (KO) and *AtNIGR1* (OE) plants had lower and higher electrolyte leakage over time than the Col-0 WT plants (Figure 3C). The current study hypothesized that the NO-downregulated *AtNIGR1* gene negatively regulates *R*-gene-mediated resistance based on the results of transcript accumulation of *AtPR1* and *AtPR2* genes and electrolyte leakage. The NO-downregulated *AtNIGR1* negatively regulates systemic acquired resistance in plants.

Plants, like other multicellular organisms, have established an intrinsic mechanism to systemically communicate the occurrence of a wound or an external stimulus to enable them to escape or defend themselves [31]. During pathogen attacks, signaling in plants is critical for triggering the required defense mechanisms to counteract the stress. However, one of the plant’s responses to pathogen infection is the production of long-term constitutive barriers known as SAR [4,32], which is characterized by the activation of a wide range of host defense mechanisms, both on a local and systemic level [32,33]. We found that after inoculating *Pst* DC3000 *avr*B, the expression of *AtPR1*, *AtPR2*, *AtG3Pdh*, and *AtAZI* genes was enhanced at all-time points in the *atnigr1* (KO) plants compared to that in the Col-0 WT and the disease-susceptible controls (*atgsnor1–3* and *atsid2* plants) (Figure 4). However, at all-time points, the transcript accumulation of *AtPR1*, *AtPR2*, *AtG3Pdh*, and *AtAZI* genes in the OE of the *AtNIGR1* gene was significantly reduced in comparison to that in the Col-0 WT (Figure 4). Therefore, the current investigation suggests that the NO-downregulated gene *AtNIGR1* negatively regulates SAR in *Arabidopsis* in response to the inoculation of *Pst* DC3000 *avr*B.

Furthermore, relying on the predicted putative protein–protein interaction network (Figure 5A and Table 1), we could say that the regulatory role of the NO-downregulated *AtNIGR1* gene might involve active interaction with many other proteins that may antagonize or work in synergy during a bacterial pathogen infection.

## 5. Conclusions

Plant responses to pathogenic attacks include activating various signaling networks and metabolic pathways to equip the plant with a robust defense system. In the process, both positive and negative regulators are either induced or suppressed, and their interaction is crucial to ensure balanced cellular activity. The current study investigated the role of the *AtNIGR1* gene in plant growth and development, as well as under oxidative and nitro-oxidative stress conditions. In addition, the involvement of the *AtNIGR1* gene in plant basal defense, *R*-gene-mediated defense, or systemic acquired resistance (SAR) was assayed. Based on the recorded enhanced resistance of *atnigr1 (*KO) and the susceptible response of the *AtNIGR1* (OE), coupled with differential transcript accumulation patterns of PR-related genes (*AtPR1* and *AtPR1*, as well as that of the SAR-related genes (*AtAZI* and *AtG3Pdh*) in the target mutant lines, *AtNIGR1* is here proposed to negatively regulate the plant basal defense, *R*-gene-mediated resistance, and SAR in response to pathogenic bacteria. To gain further insight into the role of *AtNIGR1*, additional experiments, such as in vitro protein–protein interaction studies and promoter analysis, could be conducted. These experiments would provide a more detailed understanding of the specific mechanisms by which *AtNIGR1* operates and its potential as a target for stress adaptation in plants.

## Figures and Tables

**Figure 1 antioxidants-12-00989-f001:**
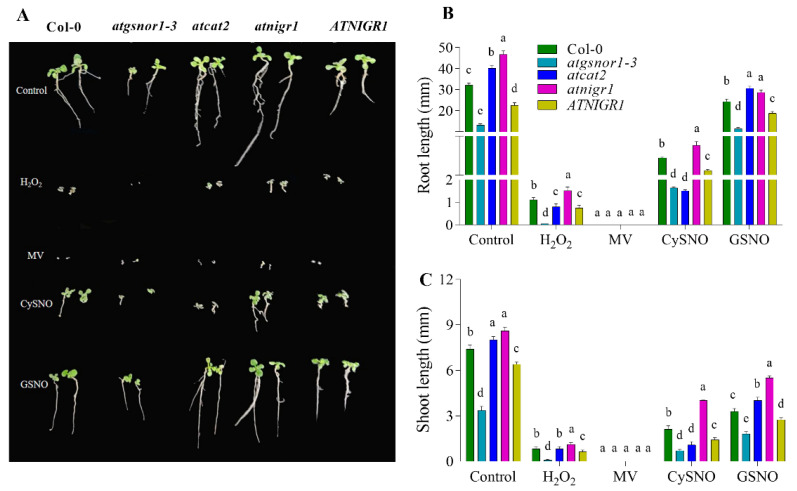
After exposure to oxidative and nitro-oxidative stress conditions, response of the *AtNIGR1* gene KO and OE plants, as well as the appropriate control plants. (**A**) Phenotypes of the indicated genotypes, (**B**) root length, and (**C**) shoot length. The mean of at least three replicates was used for all data points, and the experiment was repeated twice with nearly identical findings. The one-way analysis of variance (a, b, c, d, and e) represents the significant difference between the plants in the same treatment followed by Duncan’s multiple range test utilizing a statistical analysis system (SAS 9.1).

**Figure 2 antioxidants-12-00989-f002:**
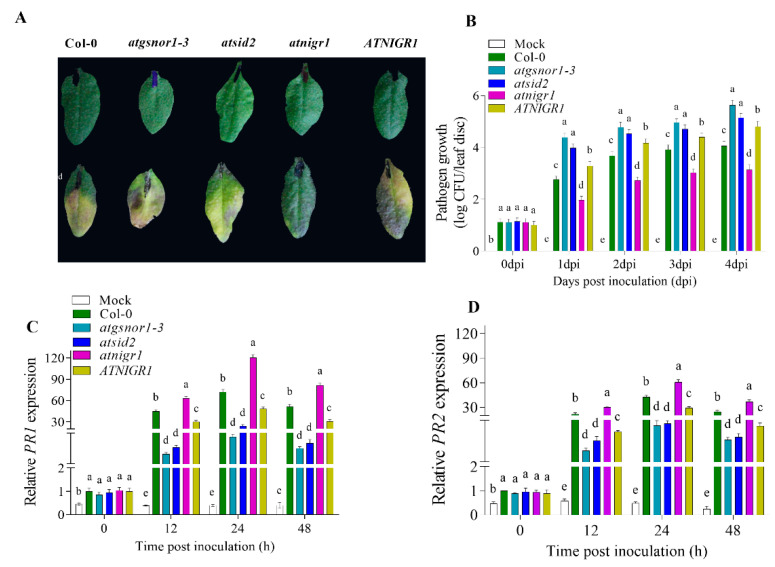
Negative regulation of plant basal defense by the *AtNIGR1* gene. (**A**) Symptoms that developed in the plants after being inoculated with *Pst* DC3000 (*vir*), (**B**) pathogen proliferation from the inoculated leaves, and relative expression of (**C**) *AtPR1* and (**D**) *AtPR2* genes. The error bars show ± the standard error (SE) of the three replicates, and all data points are the means of the three replicates. The one-way analysis of variance (a, b, c, d, and e) represents the significant difference between the plants at the same time point followed by Duncan’s multiple range test utilizing a statistical analysis system (SAS 9.1).

**Figure 3 antioxidants-12-00989-f003:**
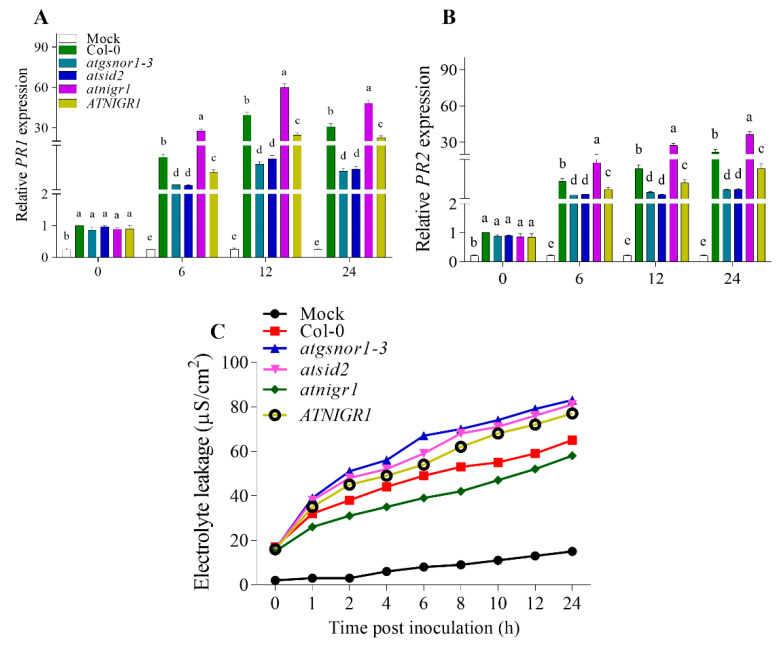
The *R*-gene-mediated resistance of the plants is negatively regulated by the *AtNIGR1* gene. After inoculation with *Pst* DC3000 *avr*B, relative expression of (**A**) *AtPR1* (**B**) *AtPR2* genes and (**C**) electrolyte leakage. The error bars show ± the standard error (SE) of the three replicates, and all data points are the means of the three replicates. The one-way analysis of variance (a, b, c, d, and e) represents the significant difference between the plants at the same time point followed by Duncan’s multiple range test utilizing a statistical analysis system (SAS 9.1).

**Figure 4 antioxidants-12-00989-f004:**
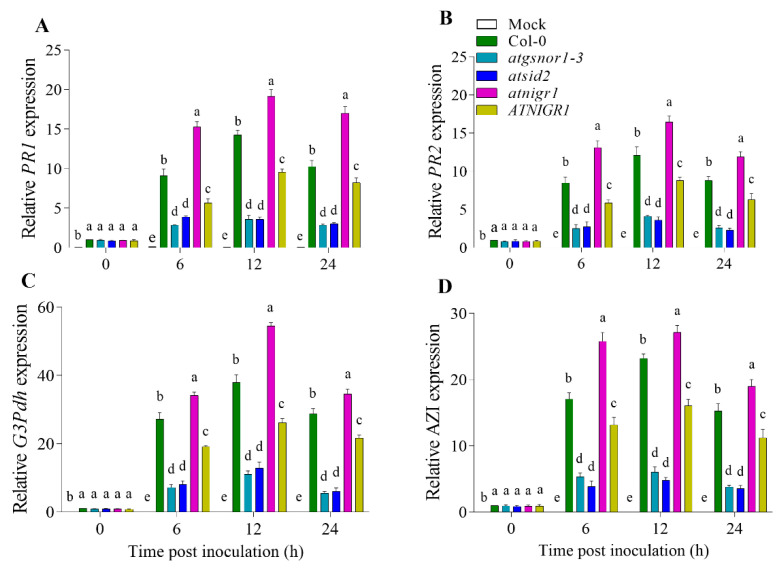
Differential regulation of SAR by the *AtNIGR1* gene. After inoculation with *Pst* DC3000 *avr*B, the relative expression of (**A**) *AtPR1*, (**B**) *AtPR2*, (**C**), *At*G3Pdh, and (**D**) *AtAZI* genes in the systemic leaves of the indicated genotypes. The error bars show ± the standard error (SE) of the three replicates, and all data points are the means of the three replicates. The one-way analysis of variance (a, b, c, d, and e) represents the significant difference between the plants at the same time point followed by Duncan’s multiple range test utilizing a statistical analysis system (SAS 9.1).

**Figure 5 antioxidants-12-00989-f005:**
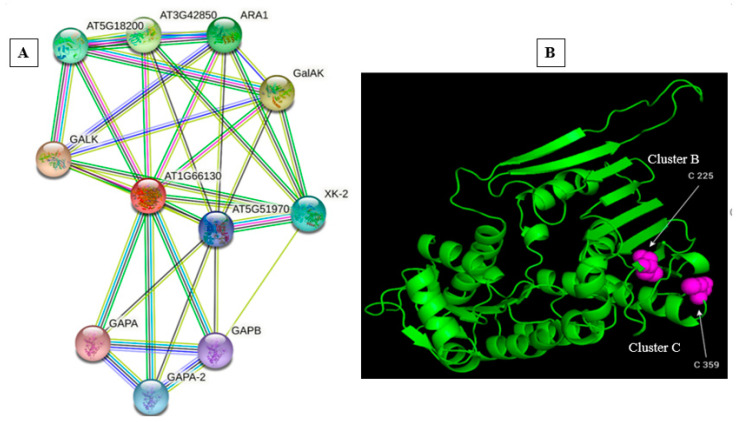
Prediction of *S*-nitrosylation-target sites and in silico protein–protein interaction. (**A**) Secondary 3D structure of AtNIGR1 protein (green colored α helices and β sheets with most exposed cysteine residues predicted as putative targets of NO-mediated S-nitrosylation to drive changes in the function of AtNIGR1 (see highlighted cysteine residues as red in table and purple in the image)). The GPS algorithm grouped prediction data into clusters B and C. (**B**) The network is the prediction of the functional protein–protein interactions (https://string.org (accessed on 7 January 2023)) between *AtNIGR1* and other identified proteins in plants. We obtained this data from the said website on 7 January 2023.

**Table 1 antioxidants-12-00989-t001:** Predicted functional partners of *AtNIGR1*.

Symbols	Full Name	Score
GALK	Mevalonate/galactokinase family protein	0.829
GalAK	Galacturonic acid kinase	0.829
AT3G42850	Mevalonate/galactokinase family protein	0.829
ARA1	Arabinose kinase; Arabinose kinase	0.829
AT5G18200	UDP glucose-hexose-1-1-phosphate uridylyltransferase	0.689
XK-2	Putative xylulose kinase	0.671
GAPA-2	Glyceraldehyde-3-phosphate dehydrogenase (NADP+) (phosphorylating)	0.653
AT5G51970	GroES-like zinc-binding alcohol dehydrogenase family protein	0.649
GAPB	Glyceraldehyde-3-phosphate dehydrogenase (NADP+) (phosphorylating)	0.639
GAPA	Glyceraldehyde-3-phosphate dehydrogenase (NADP+) (phosphorylating)	0.630

**Table 2 antioxidants-12-00989-t002:** Cysteine molecules predicted as potential targets for S-nitrosylation.

Position	Peptide	Score	Cut Off	Cluster
225	SVGTILSCTASLQFG	2.647	2.454	Cluster B
359	KKSVDIGCEVVHL	21.547	20.743	Cluster C

## Data Availability

Data is contained within the article.

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
