# Peer review of "Enhanced Resistance of atnigr1 against Pseudomonas syringae pv. tomato Suggests Negative Regulation of Plant Basal Defense and Systemic Acquired Resistance by AtNIGR1 Encoding NAD(P)-Binding Rossmann-Fold in Arabidopsis thaliana"

_antioxidants, 2023, doi:10.3390/antiox12050989_

Round 1
Reviewer 1 Report
Dear Authors,
When I secondly analyzed manuscript this time with added Figures into manuscript text, I still noticed some manuscript problems, which need to be improved/ corrected:
- The aim of the study was added, but still genes are not in italic- Please be aware: where were genes and where proteins described;
- Gene expression analyses is still based on only one reference gene - taking care about high scientific level of Antioxidant journal these analyses should be corrected to two reference genes;
- Figure 1 A are almost unreadable - Please provide better resolution and size;
- As I underlined before, there is no tissues expression pattern; There are still factual error – Authors stated “maximum expression was observed in germinating seeds, followed by an expression in the cotyledonary leaves, vegetative rosette leaves, and rosette leaves after flowering” – leaves, seeds are not plant’s tissue (!) ;
- Moreover, The big problem is with Figure 4 – Dis Authors create this figure ? or cited these expressions diagrams with data base like TAIR ? It should be clearly stated;
- Furthermore, Figure 5 needs more explanation in figure caption;
- Please add future prospects coming from obtained results to underline the importance of the results to the wider audience;
Therefore, my final decision is "major changes are needed"
Author Response
Responses to reviewer 1 comments
We are thankful to the reviewer for his thorough and positive comments. We have revised and incorporated all suggested changes accordingly. All changes and improvements in the revised version are done with track changes in the manuscript.
Reviewer Comment 1:
The aim of the study was added, but still genes are not in italic- Please be aware: where were genes and where proteins described.
Author’s Response:
We apologize for the inconvenience. We have now italicized the genes, where applicable, in the manuscript.
Reviewer Comment 2:
Gene expression analyses is still based on only one reference gene - taking care of the high scientific level of Antioxidant journal these analyses should be corrected to two reference genes.
Author’s Response:
We appreciate the concern raised by the reviewer and the interest to improve our manuscript. We would like to specify that it is commonly accepted that the expression of a target gene should be normalized to that of one of the identified housekeeping genes known to have a constitutive expression. In our case, investigated the expression of well-known pathogenesis-related genes (PR1 and PR2) and G3DPH and AZI known as system-acquired resistance (SAR) marker genes) as reference genes for their established roles in governing plants’ response to pathogens. To investigate the role of our target gene AT1G66130, we used Arabidopsis lines lacking (at1g66130) and overexpressing our target gene. Col-0 was used as the wild type, while atgsnor1-3 (high NO-producing mutant) and atsid2 (SA-deficient mutant) were used as disease susceptible controls. With these PR genes and SA or JA-related genes, and the set of control mutant lines, we believe that they help understand the involvement of our target genes in basal defense and systemic in Arabidopsis and their possible interaction.
Reviewer Comment 3:
Figure 1 A are almost unreadable - Please provide better resolution and size.
Author’s Response:
We have provided a new Figure 1 with a higher resolution as suggested.
Reviewer Comment 4:
As I underlined before, there is no tissues expression pattern; There are still factual error – Authors stated “maximum expression was observed in germinating seeds, followed by an expression in the cotyledonary leaves, vegetative rosette leaves, and rosette leaves after flowering” – leaves, seeds are not plant’s tissue (!) ;.
Author’s Response:
We agree with the reviewer’s concern and we apologize for the inconvenience. Leaves, seeds, flowers, etc. are plant organs and not tissues. We have corrected this factual error in the revised version of the manuscript.
Reviewer Comment 5:
Discussion
Moreover, The big problem is with Figure 4 – Dis Authors create this figure? or cited these expressions diagrams with data base like TAIR ? It should be clearly stated;
Author’s Response:
Figure 4 was obtained from a public repository, the eFP browser in TAIR. We have now specified the source of this figure as suggested.
Reviewer Comment 6:
Furthermore, Figure 5 needs more explanation in figure caption.
Author’s Response:
We are thankful to the reviewer for the remark. We have added more information in the caption of Figure 5.
Reviewer Comment:
Please add future prospects coming from obtained results to underline the importance of the results to the wider audience.
Author’s Response:
We have added the future prospects at the end of the conclusion section of the manuscript.
Reviewer 2 Report
This manuscript from Azzawi et al. describes the function characterization of a NAD(P)-binding Rossmann-fold superfamily gene (AT1G66130) in Arabidopsis growth, under oxidative and nitro-oxidative stresses, and immunity. The primary objective of this study is to investigate the role of AT1G66130 by assessing the impact of a knockout and an overexpression line of this gene in terms of PTI, ETI, and SAR activation upon bacteria treatment and growth under oxidative and nitro-oxidative stresses. Additionally, some protein structure and interaction analyses were made. While the authors provide convincing evidence of the role of AT1G66130 in Arabidopsis immunity and growth, the data could be presented and discussed more effectively.
Specifically in the abstract, certain aspects such as GO analysis, AT1G66130 expression and potential protein interactions, and S-nitrosylation sites are mentioned but later they are not elaborated on in the manuscript, which could be confusing for readers. Additionally, the final sentence of the abstract should summarize the importance and/or conclusions of the research findings, whereas currently, it appears to simply describe the results without any clear conclusion. Addressing these areas could improve the overall clarity and impact of the abstract.
The introduction is well-crafted, effectively providing the readers with the necessary background information on plant immunity and the crucial role of nitric oxide in plant growth, development, and immunity. However, it would be beneficial for the authors to explicitly emphasize the significance of their research and provide a clear rationale for the bioinformatic analysis conducted.
While the Material and Methods section provides sufficient details on the experimental procedures, it is crucial to disclose the specific virulent and avirulent bacterial strains utilized for testing PTI, ETI, and SAR. This information directly influences the readers' interpretation of the outcomes obtained from the bacterial inoculation assays. For instance, different bacterial strains may elicit different responses from the plant immune system, which could affect the conclusions drawn from the study. Therefore, a clear understanding of the bacterial strains used in each experiment is essential for interpreting the results presented and drawing definitive conclusions regarding the impact of AT1G66130 on PTI, ETI, and SAR activation.
The results section of the manuscript requires further attention. While each result is presented, the authors have not included the rationale for the experiment, or a brief overview of the methodology used. More feedback is provided in the specific comments below. It is also worth noting that the focus of this study is on the role of AT1G66130 in immunity and growth. However, Figure S2 showing the growth results is presented as a supplemental figure and it should be included in the main figures. Additionally, Figure 4 is not relevant to the study and adds no information of importance to this study. It should be removed.
The discussion section of the manuscript requires improvement. It should highlight the results and establish correlations with current literature. However, the authors have not provided an adequate explanation for the results they obtained. Additionally, the current discussion section appears to be a repetition of the combined results from immunity and growth assays. Furthermore, certain parts of the results are entirely neglected in the discussion, leaving questions as to why those experiments were conducted in the first place. Therefore, the authors should rewrite this section to provide a coherent explanation for all their findings and relate them to existing knowledge in the field.
The conclusion section of the manuscript should include a summary of the results and discussion, followed by the author's perspective on how this study will contribute to the current scientific knowledge. Furthermore, the authors could elaborate on the expected implications of their work moving forward and explain the relevance of conducting this research in the first place. Providing such insights would enhance the impact and overall value of the manuscript.
Specific comments:
1. The manuscript requires proper formatting of species names (both plant and bacteria), which should be italicized. Furthermore, mutant lines (e.g., atgsnor1-3, atsid2, etc.) should also be italicized.
2. Line 23 and the rest of the manuscript require proper formatting of “H2O2”. It should be changed to H2O2.
3. On line 24, it should be stated what CySNO and GSNO stands for, similarly to line 23.
4. The manuscript contains several incorrect acronyms. For example, on line 59, “PRRS” should be corrected to “PRRs” for consistency with accepted nomenclature. Additionally, on line 84, "differential gene expression" should be accompanied by the acronym "DGEs" and used consistently throughout the manuscript (e.g. line 87).
5. Line 72 to 75, this sentence is very long and not clear. You should consider rewriting to “NO's recognition as a crucial signaling molecule among RNS has led to an increased interest in NO-related research in plant bioscience (Tuteja et al. 2004). This interest can be attributed to NO's exceptional regulatory functions in both plants and animals, which earned it the title "Molecule of the Year in 1992” (Campbell and Hentschel 1992, Khan et al. 2023).”
6. Line 79, instead of “NO-regulated AtCLV1, AtCLV3…”, it should be “NO-regulated genes AtCLV1, AtCLV3…”
7. The sentence from line 83 to 87 is quite long and complex. Consider rewriting to “RNA-seq-mediated transcriptomic studies have shed light on the role of NO in biological systems. These studies have reported differential gene expression (DGE) patterns in response to 1 mM S-nitrosocysteine (CySNO), a NO donor, and have shown that NO is associated with the regulation of plant defense, abiotic stress response, hormone signaling, and other developmental processes (Hussain et al. 2016).”
8. In Materials and Methods, all the bacteria strains should be clearly identified (lines 140, 149, and 151). Additionally, on line 157, 159, 262, 362, 379, and 386 the bacteria name should be written as Pst DC3000 avrB. Please review the entire manuscript for references to virulent or avirulent bacteria, and make sure the specific bacterial strain is identified in each case.
9. The figure 3 legend mentions "Pst avirulent bacteria DC3000", but it is not clear whether this strain contains the avrB gene or not. At no point in the manuscript is it specified which virulent bacteria was used, readers are left to infer that Pst DC3000 was used. If that is the case, the claim that AT1G66130 can negatively regulate basal defense (PTI) is inaccurate. The Pst DC3000 strain is capable of delivering effectors and while Arabidopss does not recognize these effectors, the effectors can inhibit the PTI response. Thus, the assay will not show the full contribution of AT1G66130 to PTI. The commonly used strain in the field to evaluate PTI activation is Pst DC3000 hrcC, which lacks the ability to deliver effectors. Therefore, providing precise information regarding the bacterial strain used in the experiments is crucial not only to aid readers in comprehending the outcomes but also to draw accurate conclusions.
10. On line 146 the formatting of “5x105” should be corrected to “5x105”. Furthermore, the sentence from line 146 to line 150 is quite long and can be confusing to the reader. You should consider rewriting to “The bacterial strains were inoculated at a concentration of 5 x 105 colony forming units (CFU) into the abaxial side of the leaves of different plant lines, including wild-type (WT), atgsnor1-3, atsid2, and KO and OE lines of the AT1G66130 gene. The virulent bacterial strain (add name here) was used to investigate the function of AT1G66130 in PTI, while the avirulent bacterial strain (add name here) was used to evaluate the role of AT1G66130 in ETI and SAR.”
11. The figure legends require proper formatting. To enhance clarity, each panel of each figure should be described individually. For example, (A) should be accompanied by an explanation of the contents shown in that panel, followed by an explanation of subsequent panels (e.g., B, C, etc.) in the same order they appear in the figure.
12. As mentioned above, each section describing results should provide the rationale for each experiment and an overview of the methodology used. Currently, as written there is no context given for why and how each set of experiments were done. The results are stated without any frame of reference. It is recommended to include a few introductory sentences before presenting the results. For instance, on line 226, the following sentence could be added: “'To investigate the potential involvement of NO-downregulated AT1G66130 in root and shoot growth of Arabidopsis, plants were exposed to oxidative (H2O2 and MV) and nitro-oxidative (CySNO and GSNO) stress conditions. Our findings indicated that…”. It is advised to apply this approach to all the results presented in the manuscript.
13. In the Results section discussing the inhibition of roots and shoots by AT1G66130 some description about the mutant lines that are used as controls should be provided. Even though a comprehensive explanation is given in the Materials and Methods section, the results section should be self-contained and a full description of what these mutants are and why they are controls should be given. Similarly, on line 240 and 241, there is a need to clarify the controls used in the experiment.
14. On lines 239, 241, and 243, the terms "bacterial growth" and "pathogenic growth" are used. To avoid confusion among readers, it is recommended to use a consistent term throughout the manuscript. It is suggested to choose a term that best represents the intended meaning and is used commonly in the field. Moreover, the rest of the manuscript should be checked for consistency.
15. An explanation or purpose for the electrolyte leakage assay should be given in the Materials and Methods and results section. Refer to lines 266 and 363.
16. On line 279, it would be helpful to clarify that the leaves collected at the time points mentioned were not the inoculated leaves but rather the distal leaves. Moreover, it would be beneficial to explain the reasoning behind selecting AtPR1, AtPR2, AtG3DPH, and AtAZI genes for testing transcript accumulation.
17. The sentence on lines 291 and 292 is not clear.
18. The data presented between lines 291 and 318 lack an explanation of their relevance and rationale and are not discussed further in the manuscript. As a result, their significance to the study is unclear. To enhance the clarity and coherence of the manuscript, the authors may want to consider the following points:
o The functional characterization and GO annotation analysis shed light on AT1G66130's involvement in light stimulus response, nucleotide binding, and its localization in cytosol and nucleus. However, it is unclear how this information is relevant to AT1G66130's role in NO signaling or Arabidopsis growth and defense activation. The authors should provide a more detailed explanation or discussion to clarify the connection between these concepts.
o The gene structure analysis indicates that AT1G66130 possesses a canonical oxidoreductase domain, which suggests a potential role in cellular redox signaling and plant defense. The authors should explore this connection in more detail to provide further relevancy to their work.
o The tissue-specific expression pattern data does not add any value to the discussion and should be removed from the manuscript.
o The presentation of the data on predicted S-nitrosylation sites is not clear and lacks sufficient detail. Although AT1G66130 was identified from RNAseq data in response to CysNO, there is no clear explanation provided for why certain cysteine sites were highlighted and what is the relevancy. The predicted structure is also not properly labeled, making it difficult for readers to understand the meaning of "cluster B" and "cluster C."
o The protein-protein interaction analysis is based on computational predictions and the authors do not provide a clear explanation of its relevance to the study.
19. The authors should be careful to not over interpret the data they present. For example, the statement on line 356 that NO is involved in suppressing AT1G66130 to negatively regulate basal defense is not supported. While the results presented demonstrate that AT1G66130 negatively regulates basal defense, there is no data presented connecting AT1G66130’s role in defense to NO. This claim is reiterated later in the conclusion and should be adjusted to accurately reflect the findings of the study.
20. The current title of the manuscript is misleading and requires revision to better reflect the presented findings. While the study investigates the role of AT1G66130 in Arabidopsis growth and defense responses, the current title suggests a direct involvement of NO downregulation in this role, which is not shown in the results.

Author Response
Responses to reviewer 2 comments
We are thankful to the reviewer for his thorough and positive comments. We have revised and incorporated all suggested changes accordingly. All changes and improvements in the revised version are done with track changes in the manuscript.
Reviewer Comment 1:
Specifically in the abstract, certain aspects such as GO analysis, AT1G66130 expression and potential protein interactions, and S-nitrosylation sites are mentioned but later they are not elaborated on in the manuscript, which could be confusing for readers. Additionally, the final sentence of the abstract should summarize the importance and/or conclusions of the research findings, whereas currently, it appears to simply describe the results without any clear conclusion. Addressing these areas could improve the overall clarity and impact of the abstract.
Author’s Response:
We appreciate the suggestion made by the reviewer. We have made the necessary changes in the abstract.
Reviewer Comment 2:
The introduction is well-crafted, effectively providing the readers with the necessary background information on plant immunity and the crucial role of nitric oxide in plant growth, development, and immunity. However, it would be beneficial for the authors to explicitly emphasize the significance of their research and provide a clear rationale for the bioinformatic analysis conducted.
Author’s Response:
We have provided a rational for the bioinformatics in a broad context in the introduction as suggested.
Reviewer Comment 3:
While the Material and Methods section provides sufficient details on the experimental procedures, it is crucial to disclose the specific virulent and avirulent bacterial strains utilized for testing PTI, ETI, and SAR. This information directly influences the readers' interpretation of the outcomes obtained from the bacterial inoculation assays. For instance, different bacterial strains may elicit different responses from the plant immune system, which could affect the conclusions drawn from the study. Therefore, a clear understanding of the bacterial strains used in each experiment is essential for interpreting the results presented and drawing definitive conclusions regarding the impact of AT1G66130 on PTI, ETI, and SAR activation.
Author’s Response:
We have specified the name of the bacterial strain used to inoculate plants as requested in the materials and methods.
Reviewer Comment 4:
The results section of the manuscript requires further attention. While each result is presented, the authors have not included the rationale for the experiment, or a brief overview of the methodology used. More feedback is provided in the specific comments below. It is also worth noting that the focus of this study is on the role of AT1G66130 in immunity and growth. However, Figure S2 showing the growth results is presented as a supplemental figure and it should be included in the main figures. Additionally, Figure 4 is not relevant to the study and adds no information of importance to this study. It should be removed.
Author’s Response:
We appreciate the suggestion made by the reviewer. We have revised extensively the description of the results as recommended. We have also moved to the supplementary file the previous Figure 4 considering that it is regarded as a supporting figure.
Reviewer Comment 5:
The discussion section of the manuscript requires improvement. It should highlight the results and establish correlations with current literature. However, the authors have not provided an adequate explanation for the results they obtained. Additionally, the current discussion section appears to be a repetition of the combined results from immunity and growth assays. Furthermore, certain parts of the results are entirely neglected in the discussion, leaving questions as to why those experiments were conducted in the first place. Therefore, the authors should rewrite this section to provide a coherent explanation for all their findings and relate them to existing knowledge in the field.
Author’s Response:
The discussion section has been revised extensively as recommended by the reviewer.
Reviewer Comment 6:
The conclusion section of the manuscript should include a summary of the results and discussion, followed by the author's perspective on how this study will contribute to the current scientific knowledge. Furthermore, the authors could elaborate on the expected implications of their work moving forward and explain the relevance of conducting this research in the first place. Providing such insights would enhance the impact and overall value of the manuscript.
Author’s Response:
We are thankful to the reviewer for the valuable suggestion. The conclusion section has been revised and improved as suggested.
Reviewer Comment 7:
The manuscript requires proper formatting of species names (both plant and bacteria), which should be italicized. Furthermore, mutant lines (e.g., atgsnor1-3, atsid2, etc.) should also be italicized.
Author’s Response:
We have italicized the scientific names of the organisms and the genes throughout the manuscript. Thank you.
Reviewer Comment 8:
Line 23 and the rest of the manuscript require proper formatting of “H2O2”. It should be changed to H2O2.
Author’s Response:
We have done it throughout the manuscript. Thank you.
Reviewer Comment 9:
On line 24, it should be stated what CySNO and GSNO stands for, similarly to line 23.
Author’s Response:
The detailed names of CySNO and GSNO are added in the manuscript. Thank you.
Reviewer Comment 10:
The manuscript contains several incorrect acronyms. For example, on line 59, “PRRS” should be corrected to “PRRs” for consistency with accepted nomenclature. Additionally, on line 84, "differential gene expression" should be accompanied by the acronym "DGEs" and used consistently throughout the manuscript (e.g. line 87).
Author’s Response:
We have corrected these terms throughout the manuscript. Thank you.
Reviewer Comment 11:
Line 72 to 75, this sentence is very long and not clear. You should consider rewriting to “NO's recognition as a crucial signaling molecule among RNS has led to an increased interest in NO-related research in plant bioscience (Tuteja et al. 2004). This interest can be attributed to NO's exceptional regulatory functions in both plants and animals, which earned it the title "Molecule of the Year in 1992” (Campbell and Hentschel 1992, Khan et al. 2023).”
Author’s Response:
We have corrected it as recommended. Thank you.
Reviewer Comment 12:
Line 79, instead of “NO-regulated AtCLV1, AtCLV3…”, it should be “NO-regulated genes AtCLV1, AtCLV3…”
Author’s Response:
We have corrected it as recommended. Thank you.
Reviewer Comment 13:
The sentence from line 83 to 87 is quite long and complex. Consider rewriting to “RNA-seq-mediated transcriptomic studies have shed light on the role of NO in biological systems. These studies have reported differential gene expression (DGE) patterns in response to 1 mM S-nitrosocysteine (CySNO), a NO donor, and have shown that NO is associated with the regulation of plant defense, abiotic stress response, hormone signaling, and other developmental processes (Hussain et al. 2016).”
Author’s Response:
We have thankful to the reviewer for the comment. The paragraph in question has been improved accordingly in the introduction section.
Reviewer Comment 14:
In Materials and Methods, all the bacteria strains should be clearly identified (lines 140, 149, and 151). Additionally, on line 157, 159, 262, 362, 379, and 386 the bacteria name should be written as Pst DC3000 avrB. Please review the entire manuscript for references to virulent or avirulent bacteria, and make sure the specific bacterial strain is identified in each case. Author’s Response:
We appreciate the suggestion of the reviewer. We have revised this part of the manuscript accordingly.
Reviewer Comment 15:
The figure 3 legend mentions "Pst avirulent bacteria DC3000", but it is not clear whether this strain contains the avrB gene or not. At no point in the manuscript is it specified which virulent bacteria was used, readers are left to infer that Pst DC3000 was used. If that is the case, the claim that AT1G66130 can negatively regulate basal defense (PTI) is inaccurate. The Pst DC3000 strain is capable of delivering effectors and while Arabidopsis does not recognize these effectors, the effectors can inhibit the PTI response. Thus, the assay will not show the full contribution of AT1G66130 to PTI. The commonly used strain in the field to evaluate PTI activation is Pst DC3000 hrcC, which lacks the ability to deliver effectors. Therefore, providing precise information regarding the bacterial strain used in the experiments is crucial not only to aid readers in comprehending the outcomes but also to draw accurate conclusions.
Author’s Response:
We have specified the use of Pst DC3000 avrB in the caption of Figure 3. Regarding the use of Pst DC3000 vir, we would like to confirm that Pst DC3000 vir is commonly used when assessing basal defense in Arabidopsis. Kindly see the following references:
(Imran et al., 2016), (Shahid et al., 2019) and (Yun et al., 2011)
Reviewer Comment 16:
On line 146 the formatting of “5x105” should be corrected to “5x105”. Furthermore, the sentence from line 146 to line 150 is quite long and can be confusing to the reader. You should consider rewriting to “The bacterial strains were inoculated at a concentration of 5 x 105 colony forming units (CFU) into the abaxial side of the leaves of different plant lines, including wild-type (WT), atgsnor1-3, atsid2, and KO and OE lines of the AT1G66130 gene. The virulent bacterial strain (add name here) was used to investigate the function of AT1G66130 in PTI, while the avirulent bacterial strain (add name here) was used to evaluate the role of AT1G66130 in ETI and SAR.”
Author’s Response:
We have added the suggested sentences in the manuscript. Thank you.
Reviewer Comment 17:
The figure legends require proper formatting. To enhance clarity, each panel of each figure should be described individually. For example, (A) should be accompanied by an explanation of the contents shown in that panel, followed by an explanation of subsequent panels (e.g., B, C, etc.) in the same order they appear in the figure.
Author’s Response:
We have added the suggested changes in the captions. Thank you.
Reviewer Comment 18:
As mentioned above, each section describing results should provide the rationale for each experiment and an overview of the methodology used. Currently, as written there is no context given for why and how each set of experiments were done. The results are stated without any frame of reference. It is recommended to include a few introductory sentences before presenting the results. For instance, on line 226, the following sentence could be added: “'To investigate the potential involvement of NO-downregulated AT1G66130 in root and shoot growth of Arabidopsis, plants were exposed to oxidative (H2O2 and MV) and nitro-oxidative (CySNO and GSNO) stress conditions. Our findings indicated that…”. It is advised to apply this approach to all the results presented in the manuscript.
Author’s Response:
We have added the suggested sentences in the result section of the manuscript. Thank you.
Reviewer Comment 19:
In the Results section discussing the inhibition of roots and shoots by AT1G66130 some description about the mutant lines that are used as controls should be provided. Even though a comprehensive explanation is given in the Materials and Methods section, the results section should be self-contained and a full description of what these mutants are and why they are controls should be given. Similarly, on line 240 and 241, there is a need to clarify the controls used in the experiment.
Author’s Response:
The results section has been revised as per reviewer’s comments.
Reviewer Comment 20:
On lines 239, 241, and 243, the terms "bacterial growth" and "pathogenic growth" are used. To avoid confusion among readers, it is recommended to use a consistent term throughout the manuscript. It is suggested to choose a term that best represents the intended meaning and is used commonly in the field. Moreover, the rest of the manuscript should be checked for consistency.
Author’s Response:
We have corrected the pointed terms throughout the manuscript. Thank you.
Reviewer Comment 21:
An explanation or purpose for the electrolyte leakage assay should be given in the Materials and Methods and results section. Refer to lines 266 and 363.
Author’s Response:
We have provided an explanation on the electrolyte leakage as recommended. Thank you.
Reviewer Comment 22:
On line 279, it would be helpful to clarify that the leaves collected at the time points mentioned were not the inoculated leaves but rather the distal leaves. Moreover, it would be beneficial to explain the reasoning behind selecting AtPR1, AtPR2, AtG3DPH, and AtAZI genes for testing transcript accumulation.
Author’s Response:
We thank the reviewer for the comments and suggestions. We made the necessary changes as recommended. AtPR1 and AtPR2 are identified as (SA-dependent pathway marker genes), while AtAZI and AtG3DPH are known as SAR marker genes.
Reviewer Comment 23:
The sentence on lines 291 and 292 is not clear.
Author’s Response:
We have corrected the pointed sentence. Thank you.
Reviewer Comment 24:
The data presented between lines 291 and 318 lack an explanation of their relevance and rationale and are not discussed further in the manuscript. As a result, their significance to the study is unclear. To enhance the clarity and coherence of the manuscript, the authors may want to consider the following points:
o The functional characterization and GO annotation analysis shed light on AT1G66130's involvement in light stimulus response, nucleotide binding, and its localization in cytosol and nucleus. However, it is unclear how this information is relevant to AT1G66130's role in NO signaling or Arabidopsis growth and defense activation. The authors should provide a more detailed explanation or discussion to clarify the connection between these concepts.
o The gene structure analysis indicates that AT1G66130 possesses a canonical oxidoreductase domain, which suggests a potential role in cellular redox signaling and plant defense. The authors should explore this connection in more detail to provide further relevancy to their work.
o The tissue-specific expression pattern data does not add any value to the discussion and should be removed from the manuscript.
o The presentation of the data on predicted S-nitrosylation sites is not clear and lacks sufficient detail. Although AT1G66130 was identified from RNAseq data in response to CysNO, there is no clear explanation provided for why certain cysteine sites were highlighted and what is the relevancy. The predicted structure is also not properly labeled, making it difficult for readers to understand the meaning of "cluster B" and "cluster C."
- The protein-protein interaction analysis is based on computational predictions and the authors do not provide a clear explanation of its relevance to the study.
Author’s Response:
We are thankful to the reviewer for the comments and suggestions. We have revised extensively the results and discussion sections as recommended.
Reviewer Comment 24:
The authors should be careful to not over interpret the data they present. For example, the statement on line 356 that NO is involved in suppressing AT1G66130 to negatively regulate basal defense is not supported. While the results presented demonstrate that AT1G66130 negatively regulates basal defense, there is no data presented connecting AT1G66130’s role in defense to NO. This claim is reiterated later in the conclusion and should be adjusted to accurately reflect the findings of the study.
Author’s Response:
We have revised the results section and all sentences that seem to over interpret the results have been removed or modified.
Reviewer Comment 25:
The current title of the manuscript is misleading and requires revision to better reflect the presented findings. While the study investigates the role of AT1G66130 in Arabidopsis growth and defense responses, the current title suggests a direct involvement of NO downregulation in this role, which is not shown in the results.
Author’s Response:
We are thankful to the reviewer for the suggestion. We have modified the title to reflect the main finding of this study.
Imran, Q. M., Falak, N., Hussain, A., Mun, B.-G., Sharma, A., Lee, S.-U., . . . Yun, B.-W. (2016). Nitric oxide responsive heavy metal-associated gene AtHMAD1 contributes to development and disease resistance in Arabidopsis thaliana. Frontiers in plant science, 7, 1712.
Shahid, M., Imran, Q. M., Hussain, A., Khan, M., Lee, S. U., Mun, B. G., & Yun, B.-W. (2019). Comprehensive analyses of nitric oxide-induced plant stem cell-related genes in Arabidopsis thaliana. Genes, 10(3), 190.
Yun, B.-W., Feechan, A., Yin, M., Saidi, N. B. B., Le Bihan, T., Yu, M., . . . Spoel, S. H. (2011). S-nitrosylation of NADPH oxidase regulates cell death in plant immunity. nature, 478(7368), 264-268.
Reviewer 3 Report
comments:
1. please provide protein levels by western blot analysis.
2. Any synergistic effects of different treatment?
Author Response
Responses to reviewer 3 comments
We are thankful to the reviewer for his thorough and positive comments. We have revised and incorporated all suggested changes accordingly. All changes and improvements in the revised version are done with track changes in the manuscript.
`
please provide protein levels by western blot analysis.
Author’s Response 1:
We sincerely appreciate the concern raised by the reviewer to improve our manuscript. We are aware that western blot analysis would give more insight into the accumulation of AT1G66130 and its roles, as well as the possible interaction with other proteins under bacterial pathogen infection. However, due to the limitation in our laboratory, we could not perform this experiment as recommended. We are sorry for the inconvenience.
Reviewer Comment 2:
Any synergistic effects of different treatment?
Author’s Response:
We appreciate the concern raised. We have tried to describe the potential synergetic effect of some of oxidative or nitro-oxidative stress on the evaluated mutant lines. We believe that the results and discussion sections as revised provide more insights into the role of the target gene in the growth of plants as well as their response to bacterial pathogen infection.
Round 2
Reviewer 1 Report
Authors improved significanlty the manuscript;
Unfortunately Added Figure 5 {in current form] is disproportionate to others and the table is almost unreadible;
Moreover, it is discussed if Authors really add some future prospects, which I encourage;
Author Response
Responses to Reviewer 1 comments
We are thankful to the reviewer for his thorough and positive comments. We have revised and incorporated all suggested changes accordingly. All changes and improvements in the revised version are done with track changes in the manuscript.
Reviewer Comment 1:
Unfortunately Added Figure 5 {in current form] is disproportionate to others and the table is almost unreadable.
Author’s Response:
We are thankful to the reviewer for this valuable comment. We have improved the presentation of Figure 5 as suggested and presented the related data in Table 1 and Table 2 on lines 387-402.
Reviewer Comment 2:
Moreover, it is discussed if Authors really add some future prospects, which I encourage Author’s Response:
We added a statement that suggests future studies and direction in the conclusion section on lines 484-486. Thank you.

Reviewer 2 Report
See attached document

Author Response
Responses to Reviewer 2 comments
We are thankful to the reviewer for his comments to improve the quality of our manuscript. We have revised and incorporated all suggested changes accordingly. All changes and improvements in the revised version are done with track changes in the manuscript.
Abstract
Reviewer Comments:
- Line 23: it should be “Seeds” instead of “Seed”.
- Line 24: it should be "overexpression” instead of "overexpressor”.
- Line 30: Avirulent and virulent strains are not specified.
Author’s Response:
We have made the suggested changes in the abstract and specified avirulent and virulent strains of the bacteria in lines 23, 24 and 29–31. Thank you.
Introduction
Reviewer Comment: 1
Line 47: the authors removed “systemic acquired resistance” from the sentence. It should still be present since is the first time mentioned in the main text.
Author’s Response:
We have added systemic acquired resistance (SAR) in the manuscript on line 49. Thank you.
Reviewer Comment: 2
- Line 63 to line 66: the change suggested previously was not implemented.
Author’s Response:
We believe that the reviewer is talking about the rationale for using bioinformatics in this study. We have tried to give the essence of the use of bioinformatics and their contribution in providing solid background information on biological factors and their basic functions (lines 91–96). Thank you.
Material and Methods
Reviewer Comment: 1
Line 128: the proper formatting of “H2O2” was not fixed. It should be changed to H2O2.
- Line 137 to line 140: although the authors explain the which strains were used, the sentence is long and not clear. Consider changing to “The biotrophic bacterial pathogen Pseudomonas syringae pv. tomato DC3000 virulent (vir) strain [hereafter (Pst DC3000 vir)] was used to assess the basal defense while Pst DC3000 avrB avirulent strain was used to investigate R gene-mediated resistance and SAR as described earlier [23].”
Author’s Response:
We have fixed the formatting of H2O2 and added the suggested sentence in the manuscript on lines 138 and 147-153, respectively. Thank you.
Reviewer Comment: 2
- Line 146 to line 148: should be rewritten to address the comment above.
Author’s Response:
To clear the methodology section, we have added sentences from lines 159-166. Thank you.
Reviewer Comment: 3
- Line 156: instead of “avrB” change to “Pst DC3000 avrB”. Please keep it consistent throughout the manuscript.
Author’s Response:
We have added Pst DC3000 avrB on line 174 and kept it consistent throughout the manuscript. Thank you.
Reviewer Comment: 4
- Line 194 to line 199: this method describes the data collection for a phylogenetic tree that is shown in Figure S3. However, this data is not discussed at any point in the manuscript. This figure should be removed if it is not important enough to mention in the text.
Author’s Response:
We apologize for the inconvenience. We have now included a description of the results in Figure S3 in the manuscript under the title (In Silico Characterization, Gene Ontology, Predicted Organ-Specific Expression, and Homology) on lines 363-366. Thank you.
Results
Reviewer Comment: 1
- Line 227: change “to assess” to “in assessing”.
- Line 228: remove “to that”.
- Line 270: change “Figure 1” to “Figure 2”.
- Line 285 to 286: “inoculated” should not be italicized.
- Line 286 and line 389: change “avrB Pst DC3000” to “Pst DC3000 avrB”.
- Line 327 (338): Remove the dot.
- Line 332: change “Figure S” to “Figure S4”.
Author’s Response:
We are thankful to the reviewer for the valuable suggestion. We have added the suggested words and fixed the pointed terminologies and figures. Thank you.
Reviewer Comment: 2
The control mutants used on inhibition of roots and shoots by AT1G66130 was not explained in the results section as suggested.
Author’s Response:
We have added the suggested explanation about the controls on lines 251-254, 280-282, 312-314, and 336-339. Thank you.
Reviewer Comment: 3
The comments on the data presented between lines 291 and 318 (now lines 333 to 347) were not addressed as suggested. This comment also refers to figure 5 A and B, figure S2 A, B, and C, and figure S3 A and B. The data lack an explanation of their relevance and rationale and are not discussed further in the manuscript. The authors should consider the following points that were raised in the initial review of the study and were not addressed:
- The functional characterization and GO annotation analysis shed light on AT1G66130's involvement in light stimulus response, nucleotide binding, and its localization in cytosol and nucleus. However, it is unclear how this information is relevant to AT1G66130's role in NO signaling or Arabidopsis growth and defense activation. The authors should provide a more detailed explanation or discussion to clarify the connection between these concepts.
- The gene structure analysis indicates that AT1G66130 possesses a canonical oxidoreductase domain, which suggests a potential role in cellular redox signaling and plant defense. The authors should explore this connection in more detail to provide further relevancy to their work. The gene structure is presented on Figure S3 A but never mentioned in the manuscript. You should consider removing it if this data is not useful for the readers.
- The presentation of the data on predicted S-nitrosylation sites is not clear and lacks sufficient detail. Although AT1G66130 was identified from RNAseq data in response to CysNO, there is no clear explanation provided for why certain cysteine sites were highlighted and what is the relevancy. The predicted structure is also not properly labeled on figure 5A, making it difficult for readers to understand the meaning of "cluster B" and "cluster C". You should consider adding a label on the protein structure indicating where the clusters are located. Additionally, an explanation of what the clusters represent should be included in the Materials and Methods section. If the authors do not plan to discuss this data, it may be better to remove it altogether.
- The protein-protein interaction analysis is based on computational predictions and the authors do not provide a clear explanation of its relevance to the study. In addition, this data is not mentioned in the discussion section. Consider removing it.
Author’s Response:
We are thankful to the reviewer for bring out these comments that were not addressed in the previous round of revisions. We have added the suggested explanation on lines 352-355. We have added more information on the panel A and B of Figure 5 on their insights and we have added Tables for more clarity of the results 378-392. We have now included a description of the results in Figure S3 in the manuscript under the title (In Silico Characterization, Gene Ontology, Predicted Organ-Specific Expression, and Homology) on line 363-366. Thank you.
Discussion
Reviewer Comment: 1
- Line 373 to line 376: there isn’t evidence of a relationship between the amount of energy spent by the plant in innate immune system activation and involvement of At1G66130 in plant growth process. You should consider changing the sentence to: “The observed differences in root and shoot growth between the KO and OE of the AT1G66130 gene (Figure 1) suggest that this gene may play a role in regulating plant growth processes.”
Author’s Response:
We are thankful for your kind suggestions and added the suggested sentences on lines number 416-418. Thank you.
Reviewer Comment: 2
- It would be more beneficial to the reader if the discussion about the differential phenotypic responses between KO and OE was placed together with the AT1g66130 involvement in basal defense and SA.
Author’s Response:
We sincerely appreciate the suggestion made by the reviewer and we have arranged the discussion of the manuscript. Thank you.
Reviewer Comment: 3
- Again, the authors failed to discuss the data obtained from lines 333 to 347. As a result, their significance to the study is unclear. See the last comment on the results section of this document. Author’s Response:
We have added a brief discussion of the predicted protein–protein interaction results in the manuscript on lines 467-470, as suggested by the reviewer.
Conclusion
Reviewer Comment: 1
- Line 429 to 431: consider rewriting the sentence to “To gain further insight into the role of AT1G66130, additional experiments such as in vitro protein-protein interaction studies and promoter analysis could be conducted. These experiments would provide a more detailed understanding of the specific mechanisms by which AT1G66130 operates and its potential as a target for stress adaptation in plants.”.
Author’s Response:
We are thankful for your kind suggestion and we have added the suggested sentences in the conclusion section of the manuscript on lines 484-488.
Figure Legends
Reviewer Comment 13:
- Line 275 (287): Remove the comma at the end of the sentence.
- Line 276 (288): Change “virulent bacteria” to “Pst DC3000 (vir)”
- Line 298 (309): Change “Pst DC3000 avirulent bacteria” to “Pst DC3000 avrB”
Author’s Response:
We are thankful to the reviewer for the comment. We have added the suggested terminologies in the manuscript.

Reviewer 3 Report
No more comments
Author Response
Thank you.